# The Synergistic Action of Metformin and *Glycyrrhiza uralensis* Fischer Extract Alleviates Metabolic Disorders in Mice with Diet-Induced Obesity

**DOI:** 10.3390/ijms24020936

**Published:** 2023-01-04

**Authors:** Min-Kyeong Hong, Youngji Han, Hae-Jin Park, Mi-Rae Shin, Seong-Soo Roh, Eun-Young Kwon

**Affiliations:** 1Department of Food Science and Nutrition, Kyungpook National University, 1370 San-Kyuk Dong Puk-Ku, Daegu 41566, Republic of Korea; 2Center for Food and Nutritional Genomics Research, Kyungpook National University, 1370 San-Kyuk Dong Puk-Ku, Daegu 41566, Republic of Korea; 3Center for Beautiful Aging, Kyungpook National University, 1370 San-Kyuk Dong Puk-Ku, Daegu 41566, Republic of Korea; 4Raydel Research Institute, Medical Innovation Complex, Daegu 41061, Republic of Korea; 5Bio Convergence Testing Center, Daegu Haany University, 1 Haanydaero, Gyeongsan-si 38610, Republic of Korea; 6Department of Herbology, College of Korean Medicine, Daegu Haany University, 64 Gil, 25 Suseongro, Suseong-gu, Daegu 42158, Republic of Korea

**Keywords:** metformin, type 2 diabetes mellitus, obesity, *Glycyrrhiza uralensis* Fischer, combined treatment

## Abstract

Metformin, an antidiabetic drug, and *Glycyrrhiza uralensis* Fischer (GU), an oriental medicinal herb, have been reported to exert anti-obesity effects. This study investigated the synergistic action of metformin and GU in improving diet-induced obesity. Mice were fed a normal diet, a high-fat diet (HFD), or HFD + 0.015% GU water extract for 8 weeks. The HFD and GU groups were then randomly divided into two groups and fed the following diets for the next 8 weeks: HFD with 50 mg/kg metformin (HFDM) and GU with 50 mg/kg metformin (GUM). GUM prevented hepatic steatosis and adiposity by suppressing expression of mRNAs and enzyme activities related to lipogenesis in the liver and upregulating the expression of adipocyte mRNAs associated with fatty acid oxidation and lipolysis, and as a result, improved dyslipidemia. Moreover, GUM improved glucose homeostasis by inducing glucose uptake in tissues and upregulating mRNA expressions associated with glycolysis in the liver and muscle through AMP-activated protein kinase activation. GUM also improved inflammation by increasing antioxidant activity in the liver and erythrocytes and decreasing inflammatory cytokine productions. Here, we demonstrate that GU and metformin exert synergistic action in the prevention of obesity and its complications.

## 1. Introduction

Metformin, an antidiabetic drug belonging to the biguanide family, has been in use for decades. Among the biguanides developed for diabetes treatment, metformin exhibits superior tolerability and safety and has the advantage of improving hyperglycemia without causing weight gain [1]. The glucose-lowering effect of metformin is primarily because of its ability to inhibit hepatic gluconeogenesis [2]. The most common mechanism of action of metformin is the activation of the signaling kinase AMP-activated protein kinase (AMPK), which inhibits gluconeogenic gene expression and increases glucose transporter 4 expression, thereby increasing glucose uptake in the muscle [3,4]. Recently, studies have demonstrated that metformin phosphorylates the cAMP response element-binding protein (CREB), resulting in reduced gluconeogenesis-related gene expression [5] and inhibiting CREB-regulated transcription coactivator 2 (CRTC2) activity of CREB coactivator [6,7]. Another manner in which metformin improves the lipid profile is by decreasing hepatic steatosis [8]. Metformin exerts beneficial effects by lowering plasma lipid levels and attenuating hepatic steatosis through the inhibition of lipogenesis and elevation of fatty acid oxidation via AMPK activation in the liver [1]. In addition, it induces moderate weight loss in patients with obesity who are at risk of diabetes, decreases the rate of aging-related cancer, improves antioxidant activity, and reduces oxidative stress and inflammatory response [9,10]. The recommended first-line treatment approaches for prediabetes currently include lifestyle modifications and metformin [11]. However, as the duration of type 2 diabetes increases and if metformin monotherapy fails to meet or maintain glycemic control, combination therapy with other agents is often required [12,13]. Combination therapy is generally administered with drugs, such as dipeptidyl peptidase-4 inhibitors, insulin, and sulfonylureas [13,14].

*Glycyrrhiza uralensis* Fischer (GU), commonly known as licorice, has been used as a traditional medicine and natural sweetener since ancient times [15,16]. The biologically active ingredients of GU include glycyrrhizin, liquiritin, liquiritigenin, and flavones [16]. It can modulate and complement the properties of other herbal medicines, serving as a guide for several herbal medicines [17]. It also contains species-specific flavonoids that demonstrate excellent therapeutic effects on liver injury [15,18]. It has numerous pharmacological effects, such as anti-obesity, antitumor, antioxidant, anti-inflammatory, and antihypertensive effects [16,17,19,20].

In general, metformin and other antidiabetic drugs are often used in combination therapies; however, no studies have investigated the combination therapy of metformin and natural products. Therefore, we conducted two animal studies: (1) to determine the effective dose of GU and (2) to explore the synergistic action of metformin and GU on metabolism in mice with diet-induced obesity (DIO).

## 2. Results

### 2.1. Low-Dose GU Supplementation Alleviated DIO

Low-dose GU (LGU) supplementation markedly suppresses body weight and body weight gain without any alteration in food and energy intake (Figure 1A–C). Moreover, LGU supplementation significantly decreased food efficiency ratio (FER). Thus, these results indicate that food intake did not affect weight loss in the LGU group. In terms of the adipose tissue weights, LGU supplementation significantly suppressed the weights of subcutaneous, visceral, and total white adipose tissue (WAT) as compared to those in the HFD group (Figure 1D). Interestingly, LGU supplementation significantly decreased the subcutaneous WAT weight when compared to the high-dose GU (HGU) group. Similarly, morphological observation of lipid formation in epididymal WAT showed that the HFD group had a larger area of lipid formation than that of the LGU group (Figure 1E). However, HGU supplementation showed no difference compared to the HFD group (Figure 1F). LGU supplementation significantly decreased fasting plasma glucose levels and homeostatic model assessment of IR (HOMA-IR) compared to those in the HFD and HGU groups. Therefore, the effective dose of GU extract was established to be 0.015% of the diet.

### 2.2. Metformin and G. uralensis Fischer Supplementation Alleviated DIO

Before metformin treatment, the body weight significantly decreased in the GU group. After metformin treatment, the GUM and GU groups showed markedly suppressed body weight, and the GUM group showed significantly decreased FER without altering food and energy intake (Figure 2A,B). The GUM group showed marked decreases in the weights of the perirenal, retroperitoneum, mesenteric, visceral, and total WAT compared to those of the HFD group (Table 1). Moreover, morphological observations revealed that epididymal adipocyte size in the GUM group was the smallest among the HFD-based groups (Figure 2C). In the adipose tissue, the expression of lipid metabolism-related genes was regulated by HFDM, GU, and GUM supplementation (Figure 2D). The HFDM group showed significant upregulation of *Cpt1b* and *Cox8b* expression, and the GU group showed significant upregulation of *Lipe* and *Pnpla2* expression. The GUM group showed significantly upregulated *Adrb3*, *Pparα*, *Cpt1b*, *Cpt2*, *Cox8b*, *Ucp1*, *Lipe*, and *Pnpla2* expression compared to the HFD group.

### 2.3. Metformin and G. uralensis Fischer Supplementation Improved the Plasma Lipid Profiles and Adipokine Levels in the Mice with DIO

The levels of plasma TG, TC, and non-HDL-C in HFDM, GU, and GUM groups were significantly lower than those in the HFD group (Figure 3A). No significant difference was observed in plasma adiponectin levels among the HFD groups, whereas plasma leptin and resistin levels and the L:A ratio significantly decreased in the GUM group compared to those in the HFD group (Figure 3B). In addition, a comparison between GUM and HFDM groups showed that GUM supplementation significantly decreased plasma leptin levels. The levels of pro-inflammatory cytokines were significantly suppressed by metformin treatment (Figure 3C). The HFDM group showed significantly decreased plasma tumor necrosis factor-α (TNF-α) levels, whereas the GUM group showed significantly decreased interleukin-6 (IL-6) and TNF-α levels.

### 2.4. Metformin and G. uralensis Fischer Supplementation Improved the Hepatosteatosis in Mice with DIO

The liver weights did not differ significantly among the HFD groups, whereas the HFDM, GU, and GUM groups showed significantly decreased hepatic TG levels as compared to those in the HFD group (Figure 4A). In addition, the activities of hepatic lipogenic enzymes, including malic enzyme (ME), fatty acid synthase (FAS), and phosphatidate phosphohydrolase (PAP), were significantly decreased by HFDM, GU, and GUM supplementation (Figure 4B). Hepatic morphological observations and oil red O staining revealed lesser lipid formation and accumulation in the HFDM, GU, and GUM groups than those in the HFD group (Figure 4C). In the liver, the expression of lipid metabolism-related genes was regulated by HFDM, GU, and GUM supplementation (Figure 4D). The HFDM and GU groups showed a significantly downregulated expression of ADRP compared to the HFD group. The GUM group, a synergistic action group, showed significantly downregulated expression of *Fatp4*, *Srebp1c*, *Fas*, *Scd1*, *Acc2*, and *Adrp*, but significantly upregulated expression of *Pgc1α*, compared to the HFD group. Moreover, comparison between the GUM and HFDM groups showed that GUM supplementation significantly downregulated the expression of *Srebp1c*, *Fas*, and *Acc2* but significantly upregulated the expression of *Pgc1α*.

### 2.5. Metformin and G. uralensis Fischer Supplementation Improved the Hypoglycemia in Mice with DIO

Fasting blood glucose levels were significantly decreased by HFDM, GU, and GUM supplementation (Figure 5A). Moreover, the intraperitoneal glucose tolerance tests (IPGTT) (60 min) and area under the curve (AUC) were significantly decreased in the metformin treatment groups (Figure 5B), and the GUM group showed markedly decreased HOMA-IR (Figure 5C). Furthermore, hepatic glycogen content and hepatic enzymatic activity of phosphoenolpyruvate carboxykinase (PEPCK) were significantly decreased in the GUM supplement groups compared to those in the HFD group (Figure 5D).

### 2.6. Metformin and G. uralensis Fischer Supplementation Increased AMPK-Related mRNA and Protein Expression in the Mice with DIO

In the liver, epididymal WAT (eWAT), and muscle, the expression levels of the glucose metabolism-related genes were regulated by HFDM, GU, and GUM supplementation (Figure 6A). The HFDM group showed significantly upregulated hepatic *Prkaa2*, *Prkab1*, *Hk2*, and *Pdhb*; eWAT *Prkab1*; and muscle *Pprkab1* and *Glut4* expression compared to the HFD group. However, the GU group showed significantly upregulated hepatic *Pdhb*; eWAT *Prkab1* and *Glut4*; and muscle *Glut4* and *Gpi1* expression. The GUM group, the synergistic action group, showed significantly upregulated hepatic *Prkaa2*, *Prkab1*, *Hk2*, and *Pdhb*; eWAT *Prkab1* and *glut4*; and muscle *glut4*, *Hk2*, *Gpi1*, *Pkm2*, and *Aldoa* expression compared to the HFD group, while it showed significantly downregulated hepatic *G6pc*, *Pepck*, and Crtc2 expression. Consistent with these results, the protein expression of phospho-*AMPKa*, which is the activated form of the AMPK, in HFDM and GU groups was higher than the HFD group in hepatic tissue (Figure 6B). Moreover, the GUM group had the highest protein expression of phosphor-AMPKa (Thr172).

### 2.7. Metformin and G. uralensis Fischer Supplementation Improved the Activities of Erythrocyte and Hepatic Antioxidant Enzymes and Inflammation in Mice with DIO

The activities of the erythrocyte antioxidant enzyme SOD were significantly increased in the metformin-treated groups compared to those in the HFD group, and the H_2_O_2_ lipotoxicity marker level was significantly decreased in the HFDM, GU, and GUM groups (Figure 7A). In addition, the HFDM, GU, and GUM groups showed significantly increased glutathione reductase (GR) activity compared to that in the HFD group. The HFDM and GU groups showed significantly increased glutathione peroxidase (GPx) activity, and the GU and GUM groups showed significantly increased paraoxonase (PON) activity (Figure 7B). Additionally, glutathione (GSH) levels were significantly higher in the GUM group than in the HFD group. The levels of aspartate aminotransferase (AST) and alanine aminotransferase (ALT), which are indicators of hepatotoxicity, and blood urea nitrogen (BUN), which is a renal function index, showed no significant differences among the HFD groups (Figure 7C).

## 3. Discussion

Metformin is an orally administered drug that has been used for more than 60 years as a first-line antidiabetic drug either alone or in combination with other anti-hyperglycemic drugs, owing to its safety profile and favorable cardiovascular outcomes. In this study, we first re-verified the anti-obesity effect of *G. uralensis* Fischer, an oriental medicinal herb, with an antidiabetic effect, and confirmed whether the effect was observed in a dose-dependent manner. The synergistic effect of metformin and *G. uralensis* Fischer was observed. This is the first study to investigate the anti-obesity and synergistic effects of metformin and *G. uralensis* Fischer, in mice with DIO.

*G. uralensis* Fischer has been used in combination with other herbal medicines rather than as a monotherapy. Although *G. uralensis* Fischer has some anti-obesity and antioxidant properties, there have been no studies performed to evaluate its dose-dependent anti-obesity effect to define the appropriate dose of GU. In our study, LGU supplementation markedly decreased body weight and body fat. Interestingly, LGU supplementation significantly decreased subcutaneous fat mass, plasma glucose level, and HOMA-IR compared to those on HGU supplementation. In accordance with previous studies, at a high dose, bioactive compounds may lose their effectiveness and act as pro-oxidants. A high dose of epigallocatechin-3-gallate may induce its potential toxic effects, which shows that dose-dependent hepatotoxicity is correlated with increased hepatic lipid peroxidation [21]. Additionally, high doses of flavonoids generate superoxide anion radicals, and thus, the products of lipid peroxidation increase [22]. Our study is the first to suggest that LGU is more effective than HGU, similar to the observations described in previous studies.

AMPK is a central regulator of energy homeostasis that coordinates metabolic pathways and balances nutrient supply with energy demand. Metformin, a drug widely used to treat type 2 diabetes mellitus, has been shown to activate AMPK in the liver, thereby reducing gluconeogenesis and enhancing insulin sensitivity [23]. Activated AMPK in the liver upregulates *Glut4* expression by increasing glucose uptake in the muscle and adipocytes [4]. Moreover, metformin promotes glucose oxidation in the muscles [24]. The hepatic mRNA expression of *Prkab1*, an AMPK subunit, was significantly upregulated in the GUM group compared to that in the HFD group, and the expression of *Prkag1*, another subunit of AMPK, in the HFDM, GU, and GUM groups was upregulated. Furthermore, protein expression of phosphor-AMPK-a was the highest among the experimental group. Thus, *Glut4* mRNA expression was significantly upregulated in the eWATs of the GU and GUM groups compared to that in the HFD group. Moreover, the *Glut4* mRNA expression was significantly upregulated in the muscles of the HFDM, GU, and GUM groups compared to that in the HFD group. Furthermore, the GUM group showed significantly upregulated mRNA expression of glycolysis-related genes such as *Hk2*, *Gpi1*, *Pfkm*, and *Aldoa*. Supplementation with the combination of metformin and *G. uralensis* Fischer increased glycolysis in the liver and muscles and glucose uptake in the eWAT and muscle but decreased gluconeogenesis in the liver. Consistent with these results, the GUM group showed significantly decreased blood glucose and plasma glucose levels and the AUC of the IPGTT. HOMA-IR measures glucose–insulin homeostasis as a method to evaluate insulin resistance [25]. The HOMA-IR value in the HFD group was significantly higher than that in the ND group, but it was significantly lower in the GUM group than in the HFD group. Altogether, the synergistic action of metformin and *G. uralensis* Fischer supplementation increased glucose uptake in adipocytes and muscles, enhanced hepatic and muscle glycolysis with AMPK activation, and decreased hepatic gluconeogenesis. These synergistic actions of metformin and *G. uralensis* Fischer supplementation resulted in hypoglycemic effects, as evidenced by decreased plasma glucose levels.

Interestingly, in our study, the elevated hepatic glycogen level was observed in the liver along with the increased activity of PEPCK, which is a gluconeogenic enzyme, in the HFD group. In a state of energy overload caused by a high-fat diet, glycogen use as an energy source is reduced and glucose metabolism seems to be active compared to the ND group. In addition, a previous study suggested that abnormally elevated fasting glucagon levels with fasting plasma glucose levels occurred via inactivation of Akt and upregulation of FoxO1 activity [26]. Meanwhile, metformin and GU supplementation normalized the impaired glucose metabolism through the hepatic glycogen reduction and PEPCK activity.

Furthermore, AMPK regulates lipid metabolism by inhibiting fatty acid synthase-related markers and stimulating fatty acid oxidation [27]. In this study, hepatic *Prkab1*, an AMPK subunit, and phospho-AMPKa expression in the GUM group was significantly increased compared to the HFD group. Thus, the expression of genes related to fatty acid synthesis was significantly decreased, but that of genes related to fatty acid oxidation was significantly increased. Consistent with these results, the activities of hepatic lipogenic enzymes were significantly lower in the GUM group than in the HFD group. The GUM group showed significantly decreased levels of hepatic TG, whereas the levels of cholesterol and fatty acids tended to decrease compared to those in the HFD group. Moreover, hepatic morphological observations revealed smaller lipid formation in the GUM group than in the HFD group. Consequently, the synergistic action of supplementation with metformin and *G. uralensis* Fischer water extract inhibits hepatic lipid accumulation by increasing fatty acid oxidation and decreasing lipogenesis.

Obesity is associated with a state of chronic inflammation caused by increased levels of serum pro-inflammatory cytokines and decreased levels of anti-inflammatory adipokines, such as adiponectin, which may also contribute to adipose tissue inflammation [28]. In previous studies, metformin and *G. uralensis* Fischer improved obesity-related inflammation by decreasing leptin and increasing adiponectin levels, respectively [29,30]. In our study, the levels of plasma inflammatory cytokines (IL-6 and TNF-α) were significantly higher in the HFD group than in the ND group. However, the HFDM group showed significantly decreased plasma TNF-α levels, and the GUM group showed significantly decreased levels of inflammatory cytokines (IL-6 and TNF-α).

Oxidative stress induces adipokine imbalance, increases reactive oxygen species production, and reduces antioxidant activity, resulting in oxidative damage and exacerbating inflammation and injury [31,32]. Several studies have reported that metformin and licorice extract reduce oxidative stress and possess anti-inflammatory and antioxidant properties [16,33]. This study showed that the erythrocyte H_2_O_2_ content in the HFD group was significantly higher than that in the ND group. However, the metformin-treated groups, HFDM and GUM, showed significantly increased SOD activity in the erythrocytes compared to the HFD group. Therefore, erythrocyte H_2_O_2_ content was significantly lower in the GU group than in the HFD group, as well as in the HFDM and GUM groups. However, the HFDM, GU, and GUM groups showed significantly increased GR activity compared to the HFD group. The HFDM and GU groups showed significantly increased GPx activity, and the GU and GUM groups showed significantly increased PON activity. The levels of AST and ALT, which indicate the hepatotoxicity index, tended to increase in the HFD group and BUN levels were significantly higher in the HFD group than in the ND group. The HFDM, GU, and GUM groups showed decreased AST, ALT, and BUN levels. Taken together, the synergistic action of metformin and *G. uralensis* Fischer supplementation may protect against oxidative stress by improving antioxidant enzyme activity and attenuating lipid peroxidation.

Metformin combination therapy is commonly used to maintain blood glucose levels in type 2 diabetes patients. Interest in metformin combination therapy using natural product is increasing, as side effects from drug overdose are of concern. In our study, metformin with GU supplementation had synergistic effects on hyperglycemic control via AMPK activation without any liver toxicity. Metformin and GU supplementation was able to improve the metabolic status of diet-induced obesity in mice, which constitutes one of the challenged scientific findings regarding metformin combination therapy.

## 4. Materials and Methods

### 4.1. Preparation of Metformin and G. uralensis Fischer (GU) Extract

Metformin was purchased from Sigma-Aldrich (St. Louis, MO, USA), and *G. uralensis* Fischer (GU) water extract was supplied by Daegu Haany University (Gyeongsan, Korea).

GU was purchased from Bonchowon (Yeongcheon-si, Korea), and was produced according to Korean Good Manufacturing Practice (GMP). Dried GU (100 g) was extracted with 10-fold volume of boiled water at room temperature (2 h for each extraction) and then filtered. Then, the water extract was evaporated using a rotary vacuum evaporator (Sumileyela, Gyeonggi-do, Korea) at 45 °C and the solvent was evaporated in vacuo to give an extract with a yield of 11.5% by weight of GU. The prepared powder was kept at −80 °C and dissolved in water when used. Total polyphenol and flavonoid contents of the GU water extract were 28.44 ± 0.99 mg gallic acid equivalents/g and 15.55 ± 0.05 mg quercetin equivalents/g, respectively.

### 4.2. Experimental Animals and Diet

Experiment I: Evaluation of the Effective Dose of GU Extract

Male, 4-week-old C57BL/6J mice (Jabio, Suwon, Korea) were individually housed at room temperature (22 ± 2 °C) and maintained using a 12 h/12 h light/dark cycle. Twenty-one mice were randomly assigned to a high-fat diet (HFD; *n* = 7; 60 kcal% fat) group, HFD with 0.015% (*w*/*w*) *G. uralensis* Fischer (LGU; *n* = 7), or HFD with 0.03% (*w*/*w*) *G. uralensis* Fischer (HGU; *n* = 7) for 16 weeks. The diets were fed in pellet form for 16 weeks (Appendix A). The mice had free access to food and water during the experiment.

Experiment II: Synergy effect of metformin and GU combination

The experimental design is shown in Figure 8. Male, 4-week-old C57BL/6J mice (*n* = 50) were purchased from Jabio (Suwon, South Korea). The mice were housed under the same conditions as described in Experiment I. Mice were randomly assigned to a normal diet (ND; *n* = 10), HFD (*n* = 20; 60 kcal% fat), or HFD with 0.015% (*w*/*w*) *G. uralensis* Fischer (GU, *n* = 20) for 8 weeks. After 8 weeks of HFD and GU supplementation, HFD and GU groups were randomly divided into two groups for metformin treatment and fed HFD with 0.05% metformin (HFDM, *n* = 10) or HFD with 0.015% GU extract and 0.05% metformin (GUM, *n* = 10) for 8 weeks. The diets were fed in pellet form for 16 weeks (Appendix A).

The human metformin dose set by the Ministry of Food and Drug Safety guidelines is 500 mg/day for adults. This dose was converted to a mouse dose using the body surface area normalization method [34].

At the end of the diet period, the mice were sacrificed, and the blood; liver; WAT of epididymal, perirenal, retroperitoneum, subcutaneous, and interscapular depots; interscapular brown adipose tissue; and skeletal muscle samples were obtained immediately, weighed, and stored at −70 °C. The animal study protocols were approved by the Kyungpook National University Ethics Committee (approval no. KNU 2020-0090).

### 4.3. Blood Analysis

Plasma triglyceride (TG; #AM157S), total cholesterol (TC; #AM202), HDL-C (#AM203), and AST (AST; #AM103-K), ALT (ALT; #AM102) concentrations were determined using commercially available enzymatic kits (Asan, Seoul, Republic of Korea). Plasma free fatty acid (FFA; #ab65341) level was measured using enzyme-linked immunosorbent assay (ELISA) kits (Abcam, Cambridge, UK). The BUN levels were measured using a kit (#EIABUN, Thermo Fisher Scientific Inc, Millipore, MA, USA). The levels of plasma insulin, glucagon, leptin, resistin, IL-1b, IL-6, TNF-α, and monocyte chemoattractant protein (MCP)-1 were determined using a MILLIPLEX Mouse Metabolic Hormone Expanded Panel (insulin, glucagon, leptin, and resistin) and mouse cytokine/chemokine panel (IL-1b, IL-6, TNF-a, and MCP-1) kits (Merck Millipore, Billerica, MA, USA), respectively. Plasma adiponectin levels were measured using the Mouse Adiponectin/Acrp30 Quantikine ELISA Kit (#MRP300, R&D systems, Minneapolis, MN, USA). Homeostatic model assessment of IR (HOMA-IR) was calculated as (fasting glucose (mmol/L) × fasting insulin (µIU/mL))/22.5. For the intraperitoneal glucose tolerance tests (IPGTTs), mice were fasted for 12 h at 15 weeks after the start of the experiments and then injected intraperitoneally with glucose (0.5 g/kg body weight). Blood glucose levels were measured from the tail vein with a glucose analyzer (One Touch Ultra, Wayne, PA, USA) at 0, 30, 60, and 120 min after glucose injection.

### 4.4. Hepatic and Fecal Lipid Contents

The classical Folch lipid extraction method was used for lipid extraction from the liver [35]. The lipid content of the liver and feces was determined using the same enzymatic kits used for the plasma analyses.

### 4.5. Hepatic Lipid-, Glucose-, and Antioxidant-Regulating Enzyme Activities

Hepatic cytosolic, microsomal, and mitochondrial fractions were prepared according to the Hulcher and Oleson method [36], and protein levels were determined using the Bradford methods. The ME, FAS, PAP, glucokinase (GK), and PEPCK activities and the glycogen concentrations were determined as described in our previous studies [37].

Hepatic H_2_O_2_ production, GSH amount, and PON, GPx, GR, catalase, and SOD activities were measured as previously described [38].

### 4.6. Morphology of the Liver and Fat Tissues

Liver and epididymal WAT were fixed in paraformaldehyde/phosphate-buffered saline (PBS) (10% *v*/*v*) and stained with hematoxylin and eosin and oil red O; epididymal WAT paraffin-embedded sections were stained using hematoxylin and eosin [39]. For fatty liver detection, liver tissue was embedded in optimal cutting temperature (OCT) compound and stored at −80 °C [40]. The OCT-embedded samples were sectioned to obtain 4 μm thick slices and stained with oil red O for the evaluation of fat droplets. The stained areas were analyzed under an optical microscope (Nikon, Tokyo, Japan) at ×200 magnification.

### 4.7. mRNA Expression Analysis

Total RNA from all samples was reverse-transcribed to cDNA using a PrimeScript™ RT reagent Kit with gDNA Eraser (#RR047, Takara, Shiga, Japan). Real-time RT-PCR was performed using the TB Green PCR Kit (#RR820, Takara) and the CFX96 real-time system (Bio-Rad, Hercules, CA, USA). The expression level of mouse-specific GAPDH was used as an internal control. Primer sequences are shown in Table 2.

### 4.8. Western Blot

The proteins were loaded onto a 10% SDS–polyacrylamide gel, and electrophoresis was carried out in a Tris–glycine buffer for 1 h. After transferring to nylon membranes and checking the position of the bands with a Ponceau solution, the membranes were blocked (5% skim milk in TBS, 0.1% Tween-20) at room temperature for 60 min and then incubated with the primary antibody overnight at 4 °C. Each primary antibody was diluted with 5% skim milk. AMPK and GAPDH molecules were probed and detected with anti p-AMPKα (cell signaling, #2535, 1:1000, Danvers, MA, USA), anti-AMPKα (cell signaling, #5832, 1:1000, Danvers, MA, USA), and anti-GAPDH (Santa Cruz, sc-32233, 1:1000, Dallas, TX, USA). After washing, the membrane was incubated for 30 min in TBST buffer (25 mM Tris-base, 155 mM NaCl, and 0.1% Tween-20) and incubated with anti-rabbit IgG polyclonal Ab-HRP (Cell Signaling, #7074S, 1:3000, Danvers, MA, USA) or anti-mouse IgG polyclonal Ab-HRP (Cell Signaling, #7076S, 1:3000, Danvers, MA, USA) secondary antibody at room temperature for 1 h. The membrane was washed with TBST buffer for 30 min. Immunoreactive bands were developed by using an ECL kit (Pierce Chemical Co., Rockford, IL, USA), and the molecular weight of the bands was quantified by densitometry using the Image J algorithm (National Institutes of Health, Bethesda, MD, USA).

### 4.9. Statistical Analyses

Data are expressed as the mean ± standard error of the mean (SEM). Statistical analyses were performed using the SPSS software (SPSS, Inc., Chicago, IL, USA). Statistical differences between ND and HFD results were determined using the Mann–Whitney U test. The Kruskal–Wallis test was performed among the HFD groups, and Bonferroni correction was applied post hoc. Differences were considered statistically significant at *p* < 0.05.

## 5. Conclusions

The results of this study indicate that the *G. uralensis* Fischer extract showed better anti-obesity effects at the low dose compared to the high dose, and the combined administration of metformin and *G. uralensis* Fischer extract is more effective than metformin administered alone in preventing obesity and its complications, such as fatty liver, inflammation, hyperglycemia, and dyslipidemia, in mice with DIO.

## Figures and Tables

**Figure 1 ijms-24-00936-f001:**
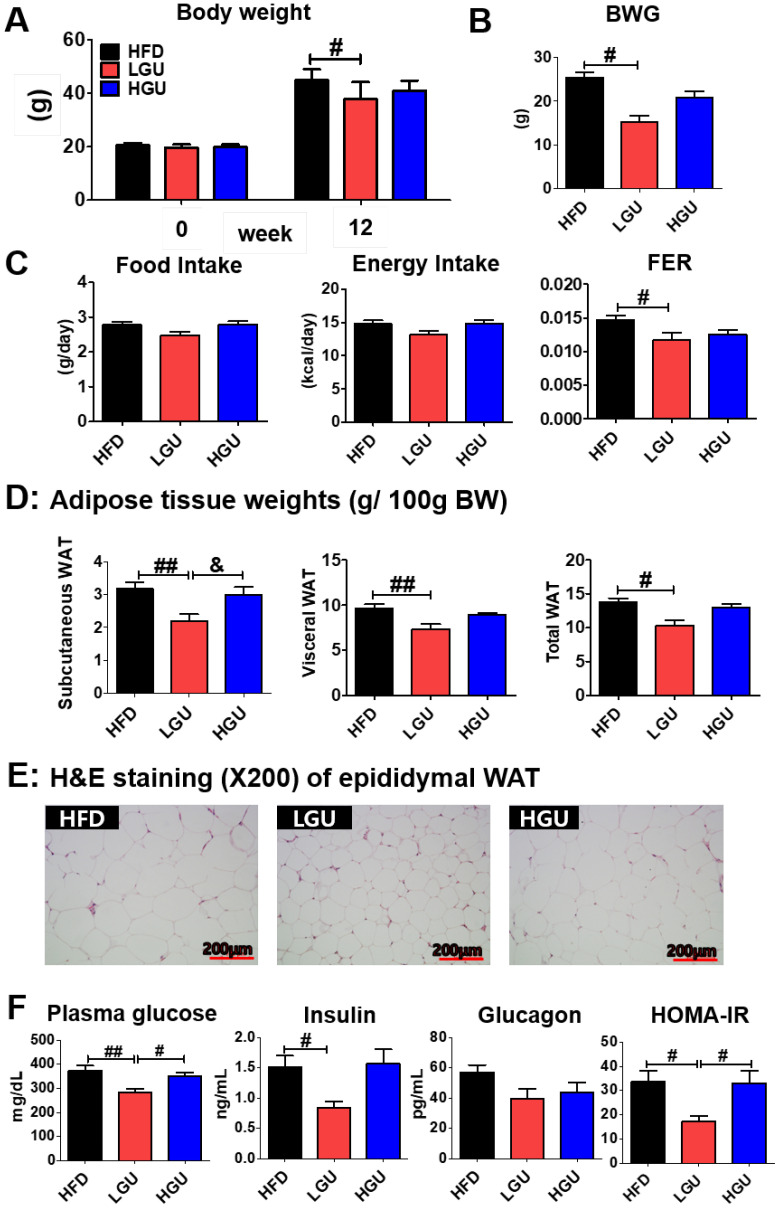
Effects of low-dose *G. uralensis* Fischer extract supplementation on diet-induced obesity with respect to body weight (**A**); body weight gain (**B**); food intake, energy intake, and food efficiency ratio (**C**); adipose tissue weight (**D**); epididymal white adipose tissue morphology (200× magnification) (**E**); and levels of plasma glucose, insulin, glucagon, and HOMA-IR (**F**). Data are presented as the mean ± SEM. Significant differences among the high-fat diet groups are indicated: # *p* < 0.05, ## *p* < 0.01. Significant differences between LGU and HGU are indicated: & *p* < 0.05. HFD, high-fat diet (60 kcal% fat); LGU, HFD + 0.015% *G. uralensis* Fischer; HGU, HFD + 0.03% *G. uralensis* Fischer.

**Figure 2 ijms-24-00936-f002:**
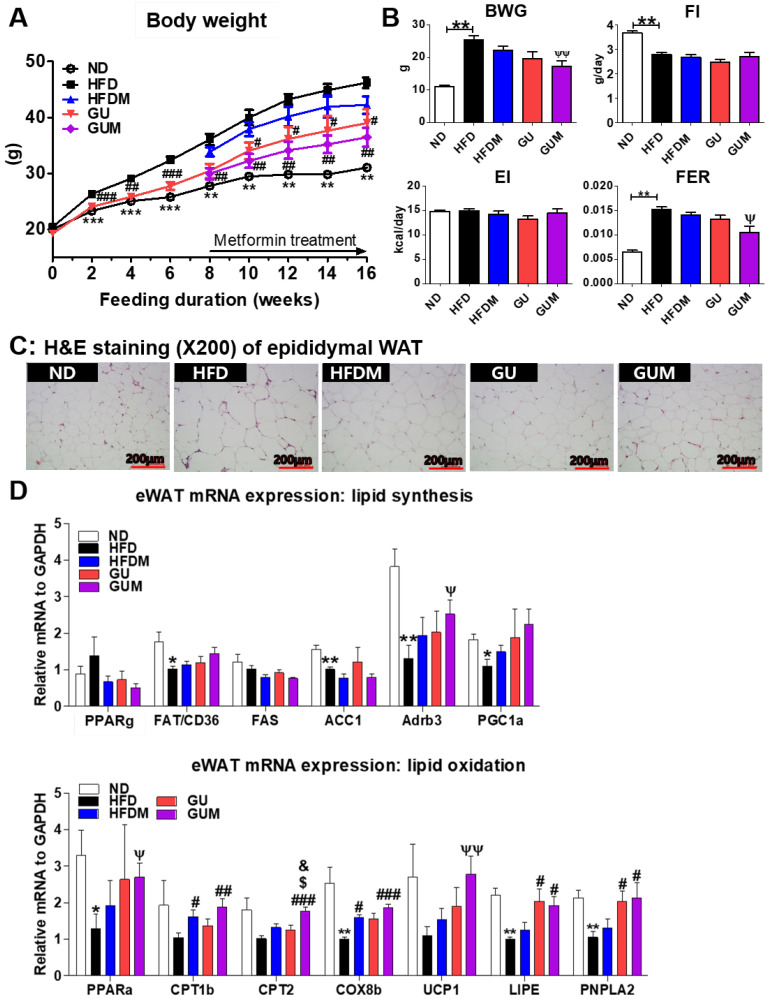
Effect of metformin, *G. uralensis* Fischer, and their combination after 16 weeks on body weight (**A**); body weight gain, food intake, energy intake, and food efficiency ratio (**B**); adipose tissue morphology (magnification 200×) (**C**); and epididymal WAT (eWAT) lipid-regulating gene expression (**D**) in C57BL/6J mice fed a high-fat diet (HFD). Data are presented as the mean ± SEM. Significant differences between HFD and ND are indicated: * *p* < 0.5, ** *p* < 0.01, *** *p* < 0.01. Significant differences among the high-fat diet groups are indicated: # *p* < 0.05, ## *p* < 0.01, ### *p* < 0.001. Significant differences between GUM and HFDM are indicated: $ *p* < 0.05. Significant differences between GUM and GU are indicated: & *p* < 0.05. When compared one-to-one, there were significant differences between GUM and HFD groups (Ψ *p* < 0.05, ΨΨ *p* < 0.01). ND, normal diet (AIN-93G 16 kcal% fat); HFD, high-fat diet (60 kcal% fat); HFDM, HFD + 50 mg/kg metformin; GU, HFD + 0.015% *G. uralensis* Fischer 0.015%; GUM, HFD + 0.015% *G. uralensis* Fischer + 50 mg/kg metformin.

**Figure 3 ijms-24-00936-f003:**
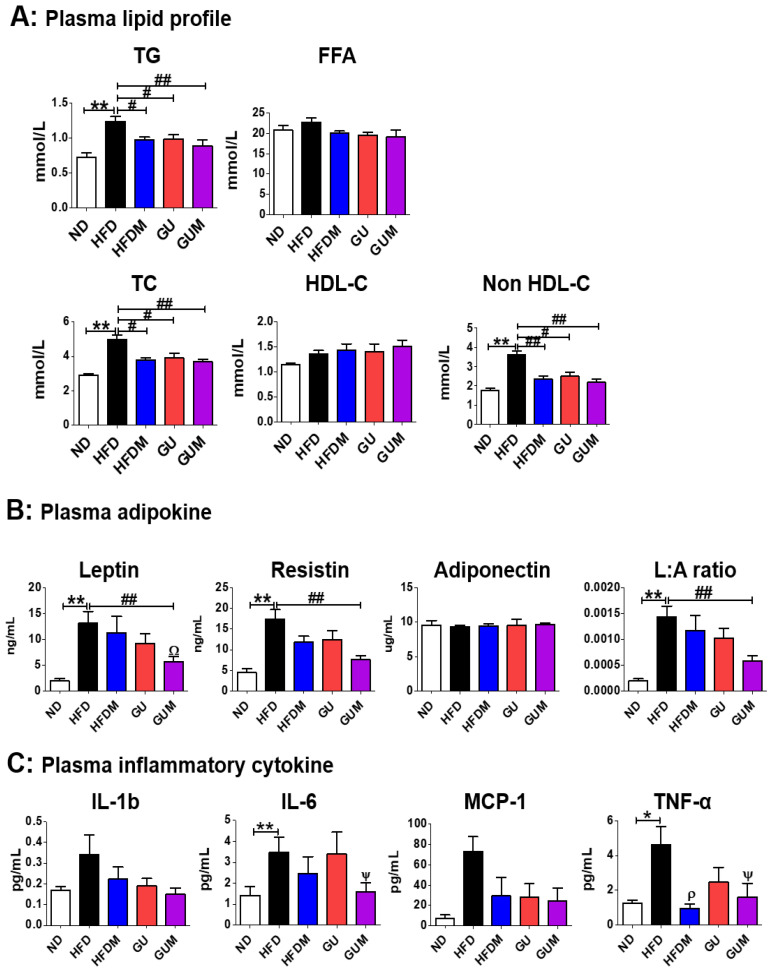
Effect of metformin, *G. uralensis* Fischer, and their combination after 16 weeks on plasma lipid profile (**A**); adipokine concentrations (**B**); and inflammatory cytokine levels (**C**) in C57BL/6J mice fed a high-fat diet. Data are presented as the mean ± SEM. Significant differences between HFD and ND groups are indicated: * *p* < 0.5, ** *p* < 0.01. Significant differences among the high-fat diet groups are indicated: # *p* < 0.05, ## *p* < 0.01. When compared one-to-one, there were significant differences between the GUM and HFD groups: Ψ *p* < 0.05. When compared one-to-one, there were significant differences between GUM and HFDM: Ω *p* < 0.05. When compared one-to-one, there were significant differences between HFDM and HFD: ρ *p* < 0.05. ND, normal diet (AIN-93G 16 kcal% fat); HFD, high-fat diet (60 kcal% fat); HFDM, HFD + metformin 50 mg/kg; GU (*G. uralensis* Fischer), HFD + GU 0.015%; GUM, HFD + GU 0.015% + metformin 50 mg/kg.

**Figure 4 ijms-24-00936-f004:**
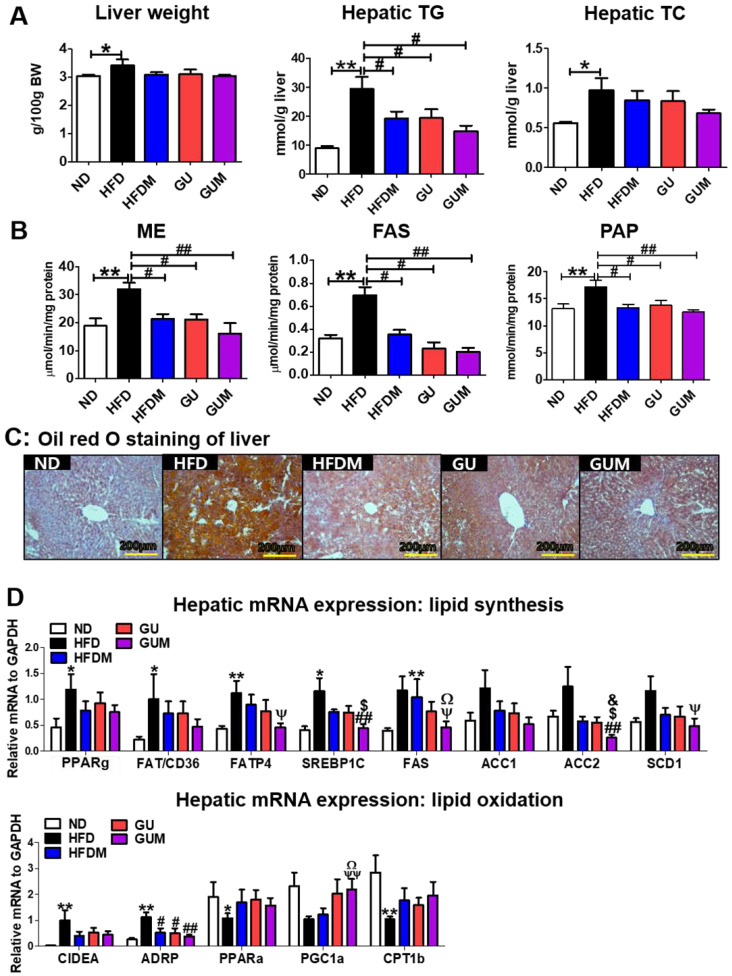
Effect of metformin, *G. uralensis* Fischer, and their combination on liver weight and hepatic lipid profiles after 16 weeks (**A**), and hepatic lipid-regulating enzyme activities (**B**), hepatic oil red O staining (magnification 200×) (**C**), and hepatic lipid metabolism-regulating gene expression (**D**) in C57BL/6J mice fed a high-fat diet (HFD). Data are presented as the mean ± SEM. Significant differences between HFD and ND are indicated: * *p* < 0.5, ** *p* < 0.01. Significant differences among the high-fat diet groups are indicated: # *p* < 0.05, ## *p* < 0.01. Significant differences between GUM and HFDM are indicated: $ *p* < 0.05. Significant differences between GUM and GU are indicated: & *p* < 0.05. When compared one-to-one, there were significant differences between GUM and HFD: Ψ *p* < 0.05, ΨΨ *p* < 0.01. When compared one-to-one, there were significant differences between GUM and HFDM: Ω *p* < 0.05. ND, normal diet (AIN-93G 16 Kcal% fat); HFD, high-fat diet (60 Kcal% fat); HFDM, HFD + metformin 50 mg/kg; GU (*G. uralensis* Fischer), HFD + GU 0.015%; GUM, HFD + GU 0.015% + metformin 50 mg/kg. *Ppar*, peroxisome proliferator-activated receptor; *Fat/cd36*, fatty acid translocase; *Fatp4*, fatty acid transporter 4; *Srebp1c*, sterol regulatory element-binding protein-1C; *Acc*, acetyl-CoA carboxylase; *Adrb3*, adrenoceptor beta 3; *Pgc1a*, peroxisome proliferator-activated receptor gamma coactivator 1-alpha; *Cpt*, carnitine palmitoyltransferase; *Cox8b*, cytochrome c oxidase subunit 8b; *Ucp1*, uncoupling protein 1; *Lipe*, lipase E; *Pnpla2*, patatin-like phospholipase domain-containing protein 2.

**Figure 5 ijms-24-00936-f005:**
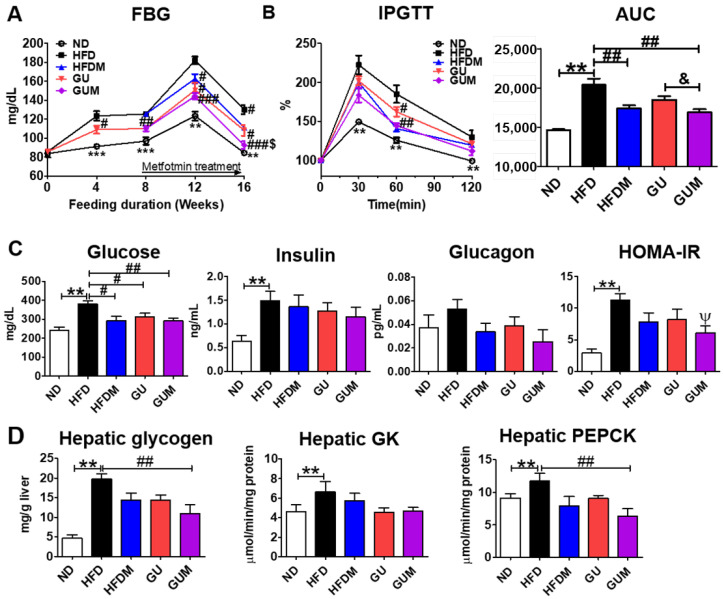
Effect of metformin, *G. uralensis* Fischer, and their combination after 16 weeks on fasting blood glucose (**A**); IPGTT and AUC (**B**); plasma glucose, insulin, and glucagon levels and HOMA-IR (**C**); and hepatic glycogen content and glucose-regulating enzyme activities (**D**) in C57BL/6J mice fed a high-fat diet. Data are presented as the mean ± SEM. Significant differences between HFD and ND are indicated: ** *p* < 0.01, *** *p* < 0.001. When compared one-to-one, there were significant differences between GUM and HFD: Ψ *p* < 0.05. Significant differences among the high-fat diet-fed groups are indicated: # *p* < 0.05, ## *p* < 0.01, ### *p* < 0.001. Significant differences between GUM and GU are indicated: & *p* < 0.05. ND, normal diet (AIN-93G 16 kcal% fat); HFD, high-fat diet (60 kcal% fat); HFDM, HFD + metformin 50 mg/kg; GU (*G. uralensis* Fischer), HFD + GU 0.015%; GUM, HFD + GU 0.015% + metformin 50 mg/kg. HOMA-IR = (fasting glucose (mmol/L) × fasting insulin (µIU/mL))/22.5.

**Figure 6 ijms-24-00936-f006:**
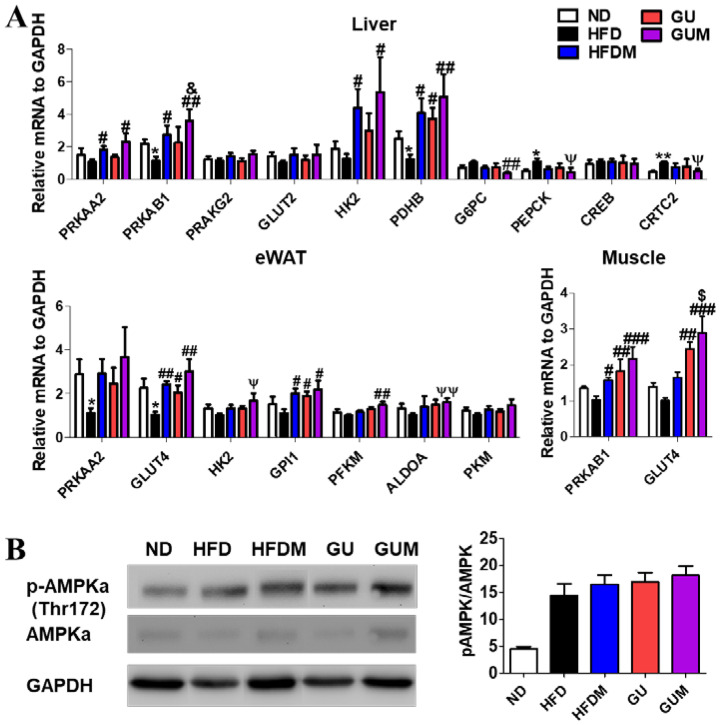
Effect of metformin, *G. uralensis* Fischer, and their combination after 16 weeks on AMPK-mediated glucose metabolism-related gene (**A**); and protein (**B**); expression in C57BL/6J mice fed a high-fat diet (HFD). Data are presented as the mean ± SEM. Significant differences between HFD and ND are indicated: * *p* < 0.5, ** *p* < 0.01. Significant differences among the high-fat diet-fed groups are indicated: # *p* < 0.05, ## *p* < 0.01, ### *p* < 0.001. Significant differences between GUM and HFDM are indicated: $ *p* < 0.05. Significant differences between GUM and GU are indicated: & *p* < 0.05. When compared one-to-one, there were significant differences between GUM and HFD groups (Ψ *p* < 0.05, ΨΨ *p* < 0.01). ND, normal diet (AIN-93G 16 kcal% fat); HFD, high-fat diet (60 kcal% fat); HFDM, HFD + metformin 50 mg/kg; GU (*G. uralensis* Fischer), HFD + GU 0.015%; GUM, HFD + GU 0.015% + metformin 50 mg/kg.

**Figure 7 ijms-24-00936-f007:**
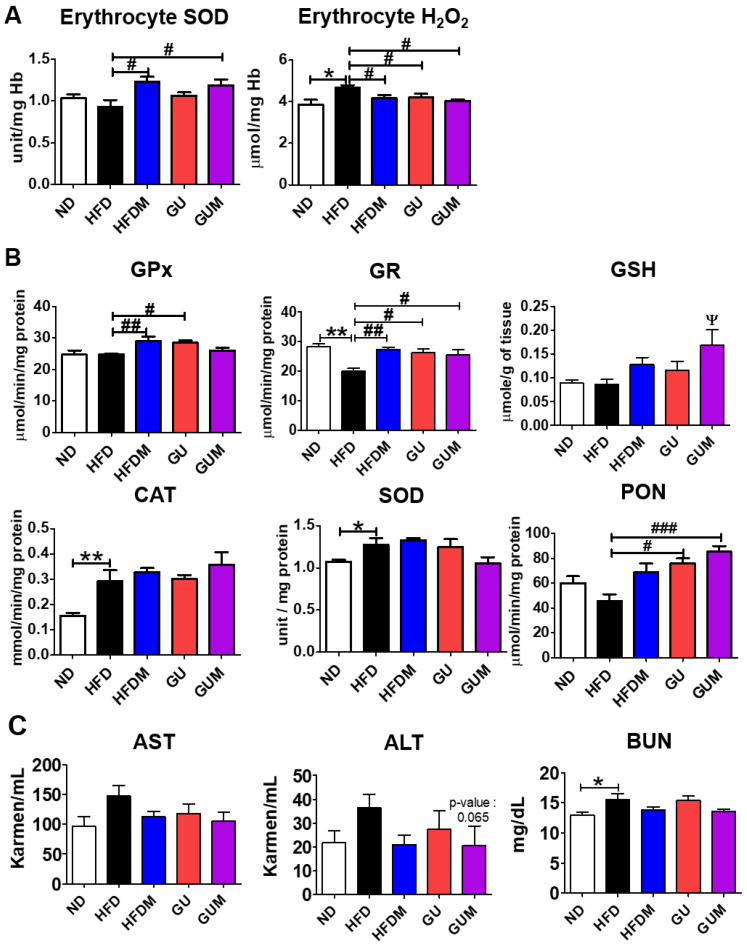
Effect of metformin, *G. uralensis* Fischer, and their combination after 16 weeks on erythrocyte H₂O₂ production and SOD activity (**A**); hepatic antioxidant enzyme activities (CAT, SOD, GPx, GR, and PON) (**B**), and plasma AST, ALT, and BUN levels (**C**) in C57BL/6J mice fed a high-fat diet. Data are presented as the mean ± SEM. Significant differences between HFD and ND are indicated: * *p* < 0.5, ** *p* < 0.01. When compared one-to-one, there were significant differences between GUM and HFD: Ψ *p* < 0.05. Significant differences among the high-fat diet groups are indicated: # *p* < 0.05, ## *p* < 0.01, ### *p* < 0.001. When compared one-to-one, there were significant differences between the HFDM and HFD groups (*p* < 0.05). ND, normal diet (AIN-93G 16 kcal% fat); HFD, high-fat diet (60 kcal% fat); HFDM, HFD + metformin 50 mg/kg; GU (*G. uralensis* Fischer), HFD + GU 0.015%; GUM, HFD + GU 0.015% + metformin 50 mg/kg.

**Figure 8 ijms-24-00936-f008:**
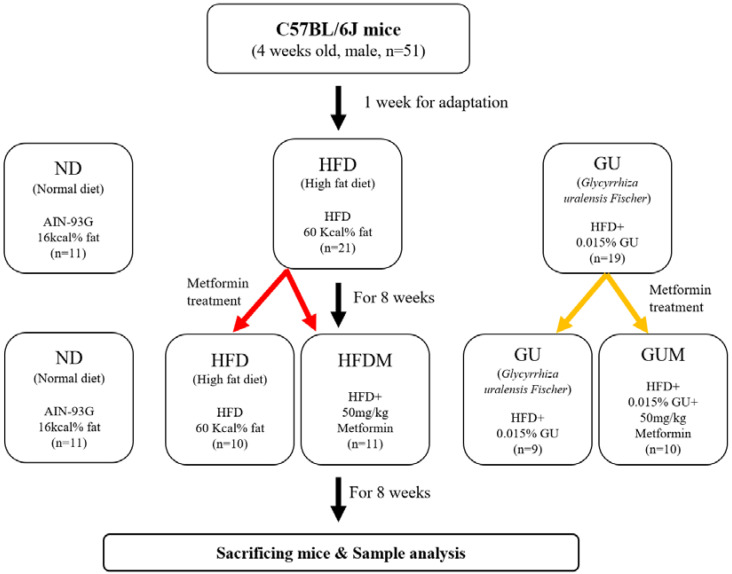
Experimental design.

**Table 1 ijms-24-00936-t001:** Effect of metformin, *G. uralensis* Fischer, and their combination after 16 weeks on adipose tissue weights in C57BL/6J mice fed a high-fat diet.

ND	HFD	HFDM	GU	GUM
g/100 g body weight				
*Epididymal white adipose tissue*				
2.31 ± 0.27	5.17 ± 0.43 **	4.89 ± 0.23	4.82 ± 0.32	4.46 ± 0.42
*Perirenal white adipose tissue*				
0.44 ± 0.1	0.95 ± 0.09 **	0.76 ± 0.07	0.64 ± 0.08 ^#^	0.5 ± 0.08 ^#^
*Retroperitoneum white adipose tissue*				
0.60 ± 0.11	1.55 ± 0.08 **	1.42 ± 0.04	1.49 ± 0.08	1.29 ± 0.06 ^##,&^
*Mesenteric white adipose tissue*				
0.69 ± 0.11	2.08 ± 0.28 **	1.36 ± 0.20	1.21 ± 0.26 ^#^	0.77 ± 0.15 ^##^
*Visceral white adipose tissue*				
4.04 ± 0.47	9.75 ± 0.50 **	8.43 ± 0.37	8.17 ± 0.57	7.02 ± 0.70 ^##^
*Subcutaneous white adipose tissue*				
1.10 ± 0.12	3.25 ± 0.22 **	2.28 ± 0.30	2.55 ± 0.30	2.08 ± 0.27 ^ΨΨ^
*Interscapular white adipose tissue*				
0.35 ± 0.1	1.00 ± 0.09 **	1.00 ± 0.12	0.92 ± 0.09	0.93 ± 0.23
*Total white adipose tissue*				
5.50 ± 0.66	14.00 ± 0.64 **	12.04 ± 0.62	11.64 ± 0.91	10.03 ± 1.01 ^##^
*Muscle*				
1.07 ± 0.03	0.86 ± 0.03 **	0.90 ± 0.03 ^#^	0.89 ± 0.04 ^#^	0.93 ± 0.05 ^##^

Data are presented as the mean ± SEM. Significant differences between HFD and ND groups are indicated: ** *p* < 0.01. Significant differences among the high-fat diet-fed groups are indicated: # *p* < 0.05, ## *p* < 0.01. Significant differences between GUM and GU are indicated: & *p* < 0.05. ΨΨ *p* < 0.01.

**Table 2 ijms-24-00936-t002:** Primer sequences of genes used for real-time PCR.

Gene	Primer Direction	Primer Sequence (5′-3′)
*Acc1*	Forward	GCC TCT TCC TGA CAA ACG AG
	Reverse	TGA CTG CCG AAA CAT CTC TG
*Acc2*	Forward	GCT GCG GTC AAG TGT ATG CG
	Reverse	CAC TGA TGC ATT TGC CCT GG
*Adrb3*	Forward	ACC AAC GTG TTC GTG ACT
	Reverse	ACA GCT AGG TAG CGG TCC
*Adrp*	Forward	GTG GAA AGG ACC AAG TCT GTG
	Reverse	GAC TCC AGC CGT TCA TAG TTG
*Cidea*	Forward	TTT CAA ACC ATG ACC GAA GTA GC
	Reverse	CCT CCA GCA CCA GCG TAA CC
*Cox8b*	Forward	TGT GGG GAT CTC AGC CAT AGT
	Reverse	AGT GGG CTA AGA CCC ATC CTG
*Cpt1b*	Forward	TGC CTT TAC ATC GTC TCC AA
	Reverse	AGA CCC CGT AGC CAT CAT C
*Cpt2*	Forward	GCC TGC TGT TGC GTG ACT G
	Reverse	TGG TGG GTA CGA TGC TGT GC
*Creb*	Forward	GAA GAA GCA GCA CGG AAG AGA
	Reverse	TCT CTT GCT GCC TCC CTG TT
*Crtc2*	Forward	ATG AAC CCT AAC CCC CAA GAC
	Reverse	CGT TCT CCT CAA TAG CAG GGA
*Fas*	Forward	GCT GCG GAA ACT TCA GGA AAT
	Reverse	AGA GAC GTG TCA CTC CTG GAC TT
*Fat/cd36*	Forward	ATT GGT CAA GCC AGC T
	Reverse	TGT AGG CTC ATC CAC TAC
*Fatp4*	Forward	CCT GGG CGA GAA CAA TGA AGT
	Reverse	ATG GGC GTG TGA TTT CCC C
*G6pc*	Forward	GGA GGA AGG ATG GAG GAA GGA ATG
	Reverse	GGT CAG CAA TCA CAG ACA CAA GG
*Gapdh*	Forward	TGC AGT GGC AAA GTG GAG AT
	Reverse	TTG AAT TTG CCG TGA GTG GA
*Glut2*	Forward	GTC AGA AGA CAA GAT CAC CGG A
	Reverse	AGG TGC ATT GAT CAC ACC GA
*Glut4*	Forward	CTG AGA ACT TAA CTG CTG AAG
	Reverse	AGG AGT TTG TTG GTG TAT TTA
*Gpi1*	Forward	CGG AAA GGT CTG CAT CAC AA
	Reverse	CCT TCA TCA GGG CCT CAG TC
*Hk2*	Forward	GAG AAC CGT GGA CTG GAC AA
	Reverse	CCA GGA AGG ACA CGT CAC AT
*Lipe*	Forward	GGC TCA CAG TTA CCA TCT CAC C
	Reverse	GAG TAC CTT GCT GTC CTG TCC
*Pdhb*	Forward	GGA GGG AAT TGA ATG TGA GG
	Reverse	CCA CAG TCA CGA GAT GAT TTG
*Pepck*	Forward	TGC CTC TCT CCA CAC CAT TGC
	Reverse	TGC CTT CCA CGA ACT TCC TCA C
*Pfkm*	Forward	GCC ATC GCC GTG TTG AC
	Reverse	GCC CTG ACG GCA GCA TT
*Pgc-1* *α*	Forward	AAG TGT GGA ACT CTC TGG AAC TG
	Reverse	GGG TTA TCT TGG TTG GCT TTA TG
*Pkm*	Forward	TTG ACT CTG CCC CCA TCA C
	Reverse	GCA GGC CCA ATG GTA CAA AT
*Pkm2*	Forward	TGC CGT GAC TCG AAA TCC C
	Reverse	GGC CAA GTT TAC ACG AAG GTC
*Pnpla2*	Forward	CAA CGC CAC TCA CAT CTA CGG
	Reverse	TCA CCA GGT TGA AGG AGG GAT
*Ppar* *α*	Forward	GCT GGA GGG TTC GTG GAG TC
	Reverse	CGG TGA GAT ACG CCC AAA TGC
*Ppar* *γ*	Forward	ATG CCA AAA ATA TCC CTG GTT TC
	Reverse	GGA GGC CAG CAT CGT GTA GA
*Prkaa2*	Forward	CAG AAG ATT CGC AGT TTA GAT GTT GT
	Reverse	ACC TCC AGA CAC ATA TTC CAT TAC C
*Prkab1*	Forward	GTT GCT GTT GCT TGT TCC AA
	Reverse	ATA CTG TGC CTG CCT CTG CT
*Prkag1*	Forward	TCT CCG CCT TAC CTG TAG TGG A
	Reverse	GCA GGG CTT TTG TCA CAG ACA C
*Scd1*	Forward	CCC CTG CGG ATC TTC CTT AT
	Reverse	AGG GTC GGC GTG TGT TTC T
*Srebp1c*	Forward	GGA GCC ATG GAT TGC ACA TT
	Reverse	CCT GTC TCA CCC CCA GCA TA
*Ucp1*	Forward	AGA TCT TCT CAG CCG GAG TTT
	Reverse	CTG TAC AGT TTC GGC AAT CCT

*Acc1/2*, acetyl-CoA carboxylase 1 and 2; *Adrb3*, adrenergic receptor beta 3; *Adrp*, adipose differentiation-related protein; *CIDEA*, cell death-inducing DFFA-like effector A; *Cox8b*, cytochrome c oxidase subunit 8b; *Cpt1b/2*, carnitine palmitoyltransferase 1b and 2; *Creb*, cAMP response element-binding protein; *Crtc2*, CREB-regulated transcription coactivator 2; *Fas,* fatty acid synthase; *Fat/cd36,* fat/cluster of differentiation 36; *Fatp4*, fatty acid transport protein 4; *G6pc*, glucose-6-phosphatase, catalytic subunit; *Gapdh*, glyceraldehyde-3-phosphate dehydrogenase; *Gck*, glucokinase; *Glut2/4*, glucose transporter 2 and 4; Gpi1, glucose phosphate isomerase 1; *Hk2*, hexokinase 2; *Lipe,* lipase; *Pdhb*, pyruvate dehydrogenase (lipoamide) beta; *Pepck*, phosphoenolpyruvate carboxykinase; *Pfkm*, 6-phosphofructokinase, muscle type; *Pgc-1α*, peroxisome proliferator-activated receptor gamma coactivator 1-alpha; *Pkm,* pyruvate kinase M1/2; *PKM2,* enzyme pyruvate kinase M2; *Pnpla2,* patatin-like phospholipase domain-containing protein 2; *Pparα/γ,* peroxisome proliferator-activated receptor alpha and gamma; *Prkaa2*, 5′-AMP-activated protein kinase subunit alpha-2; *Prkab1*, 5′-AMP-activated protein kinase subunit beta-1; *Prkag1*, 5′-AMP-activated protein kinase subunit gamma-1; *Scd1*, stearoyl-CoA desaturase-1; *Srebp1c*, sterol regulatory element binding protein 1c; *Ucp1,* uncoupling protein 1.

## Data Availability

Not applicable.

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
