# Peer review of "The Synergistic Action of Metformin and Glycyrrhiza uralensis Fischer Extract Alleviates Metabolic Disorders in Mice with Diet-Induced Obesity"

_ijms, 2023, doi:10.3390/ijms24020936_

Round 1

Reviewer 1 Report

I consider this manuscript to be well structured and developed. Moreover, the authors have carried out an exhaustive and detailed study with clearly stated results.

However, I propose some modifications that I detail below:

Lines 63-75: The authors describe in line 63 what the acronym GU corresponds to, therefore it is not necessary to use the full name in lines 65, 74 and 75.

Lines 127-132; 189-194; 240-247: I think it would be better if the full name of the genes developed in these paragraphs appeared in the "Materials and Methods" section.

Figure 3: Check the caption of the figure and the symbols of the graphs of "C".

Line 273: Insert a space.

Discussion: I think that the discussion shown is more of a summary exposition of the obtained results than a discussion. Although the study on the possible effect of G. uralensis is somewhat new, perhaps it would be good to compare it with other adjuvants used with metformin; for example, commenting on what would be the advantages of using this product over others.

Author Response

We would like to thank the reviewers for their valuable comments and feedback regarding this manuscript. We have revised the manuscript by addressing the issues raised by the reviewers. The responses to each comment are provided below.

Hong et al., studied the synergistic effects of metformin and Glycyrrhiza uralensis extract on high-fat-fed mice. The authors have reported that these compounds act synergistically and reduce hepatic steatosis.

There are a few major concerns:

  1. The age of the animals was not mentioned clearly. Whether 4 weeks old animals were fed with HFD for 16 weeks? Were the animals 20 weeks old when the tissues were collected?

--> Thank you for a kind comment. Four-week-old mice had a one-week adaptation period. They were then fed the experimental diet for 16 weeks. Therefore, the animals 21 weeks old when the tissues were collected. Added a Figure explaining the experimental design for better understanding.

  1. The primer sequences used were not given.

-->  Thank you for a kind comment. We more described the primer sequences in supplementary material.

  1. As the authors suggest that the AMPK activation may be the major contributor to the beneficial effects, whether the authors measured the active form of AMPK (phosphorylated AMPK) in the tissues? Whether the GU treatment exerts its effects by AMPK activation?

--> Thank you for your comprehensive knowledge. As your suggestion, we performed the western blot analysis to confirmed the expression of AMPKa and phosphor-AMPKa using hepatic tissue. Although we couldn’t perform the repeated experiments, phospho-AMPKa expression in GUM group was highest among the experiment. We added protein expression results and more described in our manuscript in results and discussion.

  1. The elevated hepatic glycogen level was observed in the liver along with the increased activity of gluconeogenic enzyme PEPCK in the HFD group. Does it indicate that both glycogenesis and gluconeogenesis are upregulated at the same time in HFD animals? If there are any previous reports supporting these results, kindly include them and explain.

--> Your suggestion is considerable and reasonable. In our results, PEPCK activity and hepatic glycogen levels were significantly increased simultaneously. However, it doesn’t mean that glycogen and glucose production occur simultaneously. In a state of energy overload caused by a high-fat diet, glycogen use as an energy source was reduced and glucose metabolism seems to be active compared to ND group. Meanwhile, previous study suggested that abnormally elevated fasting glucagon levels with fasting plasma glucose levels via inactivation of Akt and upregulation of FoxO1 activity. Then, we more described in our manuscript as follow: “Line 328: Interestingly, in our study, the elevated hepatic glycogen level was observed in the liver along with the increased activity of PEPCK which is a gluconeogenic enzyme in the HFD group. In a state of energy overloaded caused by a high-fat diet, glycogen use as an energy source as reduced and glucose metabolism seems to be active compared to ND group. In addition, previous study suggested that abnormally elevated fasting glucagon levels with fasting plasma glucose levels via in activation of Akt and upregulation of FoxO1 activity [40]. Meanwhile, metformin and GU supplement normalized the impaired glucose metabolism through the hepatic glycogen reduction and PEPCK activity.

  1. Provide details about HOMA-IR calculation.

--> Thank you for a kind comment. We added the HOMA-IR calculation in Figure 5 legends. Also, we already described in Line .

  1. Line 367 : “boiling water at 20-23 degrees C”? Kindly explain this.

--> Thank you for your suggestion. We totally revised the explain of GU extract.

  1. Kindly check spelling and grammar throughout the manuscript.

--> Thank you for your suggestion.

Reviewer 2 Report

Hong et al., studied the synergistic effects of metformin and Glycyrrhiza uralensis extract on high-fat-fed mice. The authors have reported that these compounds act synergistically and reduce hepatic steatosis.

There are a few major concerns:

1. The age of the animals was not mentioned clearly. Whether 4 weeks old animals were fed with HFD for 16 weeks? Were the animals 20 weeks old when the tissues were collected?

2. The primer sequences used were not given.

3. As the authors suggest that the AMPK activation may be the major contributor to the beneficial effects, whether the authors measured the active form of AMPK (phosphorylated AMPK) in the tissues? Whether the GU treatment exerts its effects by AMPK activation?

4. The elevated hepatic glycogen level was observed in the liver along with the increased activity of gluconeogenic enzyme PEPCK in the HFD group. Does it indicate that both glycogenesis and gluconeogenesis are upregulated at the same time in HFD animals? If there are any previous reports supporting these results, kindly include them and explain.

5. Provide details about HOMA-IR calculation.

6. Line 367 : “boiling water at 20-23 degrees C”? Kindly explain this.

7. Kindly check spelling and grammar throughout the manuscript.

Author Response

We would like to thank the reviewers for their valuable comments and feedback regarding this manuscript. We have revised the manuscript by addressing the issues raised by the reviewers. The responses to each comment are provided below.

Hong et al., studied the synergistic effects of metformin and Glycyrrhiza uralensis extract on high-fat-fed mice. The authors have reported that these compounds act synergistically and reduce hepatic steatosis.

There are a few major concerns:

  1. The age of the animals was not mentioned clearly. Whether 4 weeks old animals were fed with HFD for 16 weeks? Were the animals 20 weeks old when the tissues were collected?
  • Thank you for a kind comment. Four-week-old mice had a one-week adaptation period. They were then fed the experimental diet for 16 weeks. Therefore, the animals 21 weeks old when the tissues were collected. Added a Figure explaining the experimental design for better understanding.
  1. The primer sequences used were not given.
  • Thank you for a kind comment. We more described the primer sequences in supplementary material.

  1. As the authors suggest that the AMPK activation may be the major contributor to the beneficial effects, whether the authors measured the active form of AMPK (phosphorylated AMPK) in the tissues? Whether the GU treatment exerts its effects by AMPK activation?
  • Thank you for your comprehensive knowledge. As your suggestion, we performed the western blot analysis to confirmed the expression of AMPKa and phosphor-AMPKa using hepatic tissue. Although we couldn’t perform the repeated experiments, phospho-AMPKa expression in GUM group was highest among the experiment. We added protein expression results and more described in our manuscript in results and discussion.
  1. The elevated hepatic glycogen level was observed in the liver along with the increased activity of gluconeogenic enzyme PEPCK in the HFD group. Does it indicate that both glycogenesis and gluconeogenesis are upregulated at the same time in HFD animals? If there are any previous reports supporting these results, kindly include them and explain.
  • Your suggestion is considerable and reasonable. In our results, PEPCK activity and hepatic glycogen levels were significantly increased simultaneously. However, it doesn’t mean that glycogen and glucose production occur simultaneously. In a state of energy overload caused by a high-fat diet, glycogen use as an energy source was reduced and glucose metabolism seems to be active compared to ND group. Meanwhile, previous study suggested that abnormally elevated fasting glucagon levels with fasting plasma glucose levels via inactivation of Akt and upregulation of FoxO1 activity. Then, we more described in our manuscript as follow: “Line 328: Interestingly, in our study, the elevated hepatic glycogen level was observed in the liver along with the increased activity of PEPCK which is a gluconeogenic enzyme in the HFD group. In a state of energy overloaded caused by a high-fat diet, glycogen use as an energy source as reduced and glucose metabolism seems to be active compared to ND group. In addition, previous study suggested that abnormally elevated fasting glucagon levels with fasting plasma glucose levels via in activation of Akt and upregulation of FoxO1 activity [40]. Meanwhile, metformin and GU supplement normalized the impaired glucose metabolism through the hepatic glycogen reduction and PEPCK activity.

  1. Provide details about HOMA-IR calculation.
  • Thank you for a kind comment. We added the HOMA-IR calculation in Figure 5 legends. Also, we already described in Line .
  1. Line 367 : “boiling water at 20-23 degrees C”? Kindly explain this.
  • Thank you for your suggestion. We totally revised the explain of GU extract.
  1. Kindly check spelling and grammar throughout the manuscript.
  • Thank you for your suggestion.

Round 2

Reviewer 2 Report

The authors have made the necessary changes. Some minor changes are required.

1.       Kindly follow a uniform pattern for gene symbols. Mouse gene names should be italicized all over the text. The first letter of the gene name should be capitalized, and other letters should be in lowercase (Ex: Glut4).

2.       Figure 2D: eWAT mRNA expression: lipid synthesis – In the x-axis gene name “PPARr” should be Pparg

3.       Line 222: Muscle Glut4 and “g1” expression? Kindly correct it.

4.       Figure 6B: Kindly provide the phosphorylation site of the p-AMPKa measured. Error bars are missing in the bar graph.

5.       Materials and methods: Kindly provide details for western blot and measurement of activities of enzymes GSH, SOD.

Author Response

We really appreciated your reviewing. Your efforts make our manuscript flourished. 

  1. Kindly follow a uniform pattern for gene symbols. Mouse gene names should be italicized all over the text. The first letter of the gene name should be capitalized, and other letters should be in lowercase (Ex: Glut4).
  • Thank you for a kind comment. We revised the gene symbols in our manuscript.
  1. Figure 2D: eWAT mRNA expression: lipid synthesis – In the x-axis gene name “PPARr” should be Pparg
  • Thank you for a kind comment. We revised the gene name “”PPARr” to “Pparg” in Figure 2D and Figure 4D.

  1. Line 222: Muscle Glut4 and “g1” expression? Kindly correct it.
  • Thank you for a kind comment. revised the our manuscript as follow: and muscle PPRKAB1 and GLUT4 and GPI1 expression compared to that in the HFD group (Line 225).

  1. Figure 6B: Kindly provide the phosphorylation site of the p-AMPKa measured. Error bars are missing in the bar graph
  • Thank you for a kind comment. We added the phosphorylation site of the p-AMPKa (Thr172) and error bars in Figure 6B.
  1. Materials and methods: Kindly provide details for western blot and measurement of activities of enzymes GSH, SOD.
  • Thank you for a kind comment. We added the method of our study.
